# Health-Related Quality of Life Assessed in Children with Adenoid Hypertrophy

**DOI:** 10.3390/ijerph18178935

**Published:** 2021-08-25

**Authors:** Artur Niedzielski, Lechosław Paweł Chmielik, Anna Kasprzyk, Tomasz Stankiewicz, Grażyna Mielnik-Niedzielska

**Affiliations:** 1Department of Pediatric Otolaryngology, Centre of Postgraduate Medical Education, 01-813 Warsaw, Poland; arturniedzielski@wp.pl (A.N.); anna.kasprzyk@g.pl (A.K.); 2Department of Pediatric ENT, The Hospital’s Pediatric in Dziekanow Lesny, 05-092 Dziekanów Leśny, Poland; 3Independent Otoneurological Laboratory, Medical Uniwersytety of Lublin, 20-093 Lublin, Poland; tstankiewicz@o2.pl; 4Department of Pediatric Otolaryngology, Medical Uniwersytety of Lublin, 20-093 Lublin, Poland; grazyna.niedzielska@wp.pl

**Keywords:** health quality of life, adenoid hypertrophy, CHQ-PF-50, paediatric

## Abstract

Introduction: The quality of life issue began to be earnestly studied in the second half of the 20th century. It had originally been used as a criterion for measuring levels of human development in the USA and Western Europe. At first, only objective parameters were assessed, such as material goods; however, later, subjective and non-material parameters were added, such as health, freedom, and happiness. Over time, more and more attention has been paid to the subjective parameters regarding any quality of life assessment. Adenoids are physiological clusters of lymphoid tissue included in Waldeyer’s ring, which play an important role in shaping and directing the child’s local and systemic lines of defence. Adenoid hypertrophy occurs due to a variety of factors, such as recurring or chronic infections of the upper respiratory tract. Study aim: To assess health status in children with adenoid system hypertrophy compared with a group of healthy children. Materials and methods: The study group consisted of children suffering from adenoid hypertrophy, this being the most common chronic disease of the upper respiratory tract. The control group was composed of children attending nursery school (kindergarten), primary school, middle school, and high school. The study was performed by using the Child Health Questionnaire—Parent Form 50 CHQ-PF-50 (CHQ-PF50), which is a general purpose research tool based on psychometric testing when assessing physical and mental well-being in children aged 5 to 18 years. Results: There were 101 filled out questionnaires for the test group (54 girls and 47 boys), mean age 8.62 years (ranging 5–17), whilst 102 questionnaires for the controls (50 girls and 52 boys), mean age 10.58 years (ranging 5–18). Insignificant differences were found between social functioning resulting from behaviour or emotional state (REB), pain and discomfort (BP), and family cohesion (FC). Conclusions: Children suffering from adenoid hypertrophy demonstrate the largest decreases in wellbeing in the following areas: behaviour, general perception of health, and mental health.

## 1. Introduction

Quality of life can be simply defined as an area of human life that directly concerns a given person and is important to her/him [1]. A suitable questionnaire is selected in order to fullfil the study aims. General purpose questionnaires can be used when studying large populations with a variety of pathologies. It is thus possible to compare results with each other, irrespective of whether subjects are healthy or suffering from any ailments, or if subject groups are numerically different. Nevertheless, general purpose questionnaires are not useful tools for assessing discrete changes occurring in any given individual.

Specific HRQL questionnaires have been created for any specific topic that are under study. They are more sensitive in detecting deviations occurring over time and are thereby used for investigating how effective treatments are or the progression of lesions. They are not, however, suitable for evaluating patients with several diseases. Adenoids in children are physiological clusters of lymphoid tissue included in Waldeyer’s ring, which has an important role in shaping and directing local and systemic lines of immune defence. Tonsil hypertrophy arises from various factors, such as recurrent or chronic infections of the upper respiratory tract or in predisposed individuals [2,3].

The adenoid is located within the nasopharynx where there are 6–8 folds separated by furrows present on its surface. Adenoid hypertrophy may be physiological (and reversible whenever a pathological agent has ceased to function), or pathological in which case the condition remains constant [4]. The adenoid–palatine space becomes narrowed in the first phase of hypertrophy by the posterior–lower part of the pharyngeal tonsil. This is followed by an anterior hypertrophy and closure of the adenoid–choanal space [5]. Such adenoid hypertrophy cases had impaired nasal patency, changes to breathing by the mouth (adenoid face), malocclusion, voice timbre changes, nighttime snoring and night apnea. Obstruction of the Eustachian tube may lead to exudative otitis media, which in turn may lead to hearing loss [6,7]. Palpation of the pharyngeal tonsil in children has now passed into history because this examination was found to be significantly subjective when assessing the extent of tonsil hypertrophy and also because it was poorly tolerated by patients.

Adenoid hypertrophy should be diagnosed on the basis of a characteristic history and physical examination, and should also be confirmed by additional examination. At present, adenoid hypertrophy and the extent of nasopharyngeal narrowing are assessed by radiography (i.e., lateral x-ray of the nasopharynx or computed tomography of the paranasal sinuses, including an assessment of the nasopharynx). It should be noted that a suspicion of adenoid hypertrophy is not in itself an indication to perform computed tomography of the sinuses. Computed tomography constitutes an element in diagnosing chronic catarrh of the upper respiratory tract or an impaired patency of the upper respiratory tract. At present, nasal endoscopy represents the first choice in the evaluation of adenoid hypertrophy, replacing radiology [8]. Assessment of adenoid hypertrophy was based on the ratio of tonsil size to the posterior nostrils and their narrowing in the study test group, expressed as a percentage. Three degree categories for hypertrophy were used; mild <60% (without clinical symptoms), moderate (60–80%), and severe (80–100%).

## 2. Study Aim

To assess health status in children with adenoidal system hypertrophy, compared to healthy children controls. 

## 3. Materials and Methods

### 3.1. Study Materials

Subjects were children suffering from the most common chronic diseases of the upper respiratory tract, i.e., adenoid hypertrophy; inclusion criteria being ages 5 to 18 years and the presence of a chronic disease such as adenoid hypertrophy. Exclusion criteria were ages under 5 or over 18 years, acute childhood diseases, chronic childhood diseases (also included other chronic diseases such as allergic diseases, chronic sinusitis, and nasal septum deviation), and incompletely filled-in questionnaires. The control group was recruited from children attending nursery school, primary school, middle school, and high school in Warsaw and its regions. Such educational institutions and children were randomly selected. Inclusion criteria for the controls were likewise ages 5 to 18 years, whereas exclusion criteria were ages under 5 and over 18 years, acute childhood diseases, chronic childhood diseases (including tonsil hypertrophy), and an incompletely filled-in questionnaire. In all, the study consisted of 101 children with chronic adenoid disease and 102 healthy controls.

The subject test group was sub-divided into those with adenoid hypertrophy, palatine adenoid hypertrophy, and adenoid and pharyngeal hypertrophy, each in relatively small numbers. These subgroups were therefore combined into one ‘adenoid hypertrophy’ group in order to be statistically meaningful for subsequent analysis.

### 3.2. Study Methods

A Child Health Questionnaire was used—Parent Form 50 CHQ-PF-50; CHQ-PF50—which is a general purpose research instrument based on psychometric tests for assessing physical and mental well-being in children aged 5 to 18 years. It has been employed for measuring health-related quality of life, both in healthy and sick children since 1994, when it was introduced by JM Landgraff and JE Ware [9]. When constructing the questionnaire, we assumed that health status is assessed in various fields: physical and mental well-being (which includes areas relating to emotions, behaviour, and social contacts). The questionnaire actually consists of 13 categories comprising of 50 questions answered by parents/legal guardians. 

The assessment’s duration depended on the question groups. There was, however, no specific time frame regarding general perceptions of health and family cohesion. 

The current state of health was compared to that from the previous year. The preceding four weeks were investigated in the remaining groups of questions. Each answer was expressed as an appropriate numerical value. Results were calculated by the following algorithm: Sum of the obtained values divided by the number of questions answered, from which the lowest value is subtracted. This result was then divided by the range of possible outcomes, giving scores ranging between 0 and 100 [10]. The higher the score, the better the life-functioning and well-being. 

### 3.3. Statistical Analysis

This was performed on the STATISTICA package (StatSoft, Tulsa, OK, USA). A level of *p* = 0.05 was adopted as being significant and values of *p* < 0.05 are marked in bold (Table 3).

### 3.4. Variables

These were all described in the questionnaire and can be divided into two main groups: discrete and continuous. Discrete variables were further divided into those variables with two-point distributions and ones with n-point distributions. For each discrete variable, the counts and structure indices were calculated. Basic summary statistics were calculated for each continuous variable: count, arithmetic mean, standard deviation, minimum value, maximum value, skewness, and kurtosis indices, as well as positional statistics: median, Q25, Q75. Most of the continuous variables deviated from the normal distribution, which thereby affected the choice of analysis methods. Non-parametric methods were mainly used for performing the statistical tests; however, sometimes, parametric tests were also employed as part of data mining. 

Continuous Variables: These were subjected to analyses of medians, averages, and correlation. The non-parametric tests used were the Mann–Whitney test with corrections for associated ranks, the Kruskall–Wallis test using multiple tests to compare mean ranks, and the median test. Two parametric tests were used, depending on circumstances: the Student’s *t*-test or one-way analysis of variance. The first was used when the grouping variable had a two-point distribution. Equality of variance was tested by the F-test. If the grouping variable had a different distribution, analysis of variance (ANOVA) was used. Equality of variance was checked by the Brown–Forsythe test. The RIR–Tukey test was used for multiple comparisons. Spearman’s rank correlation coefficient and Tukey’s correlation coefficient were used to calculate the correlation. 

Discrete Variables: The independence analysis used was based on the chi-square test of independence. A two-tailed exact test was used in the case of four-field tables whenever numbers were smaller than expected. Appropriate groupings were used for tables with more fields. For the purposes of interpretation, Wanke’s surplus values were also calculated in the contingency tables. 

## 4. Results

The 640 CHQ-PF-50 questionnaires were distributed among the parents of randomly selected children in the control group. There were 102 questionnaires (15.93%) that met the admission criteria. Controls consisted of 50 girls and 52 boys with a mean age of 10.58 years, the youngest child being 5 years old and the oldest 18 years (Table 1). One hundred and fifty CHQ-PF-50 questionnaires were given to the parents of the test study group of children suffering from adenoid hypertrophy. There were 101 questionnaires that met the admission criteria (67.33%). This group consisted of 54 girls and 47 boys with an average age of 8.62 years; the youngest child being 5 years old and the oldest 17 years (Table 2). There were no statistically significant differences between the study group and the controls regarding age and sex.

For continuous variables in the control group, the mean values ranged from 3.78 to 97.11, the standard deviation was between 0.86 and 14.21 and the median ranged from 4.40 to 100.00. 

For continuous variables in the children with tonsil hypertrophy, mean values ranged from 3.58 to 92.63, standard deviation was between 0.76 and 14.21, and the median ranged from 3.40 to 100.00. All children in the study test group had either moderate or severe adenoid hypertrophy. There were no statistically significant relationships between the degree of tonsil hypertrophy with reductions in any of the areas examined on the quality of life. This may have been related to the type of test we used (a general purpose questionnaire).

A comparison of children suffering from adenoid hypertrophy (test group) with controls to pinpoint statistically significant differences for individual quality of life issues is shown in Table 3.

There were no statistically significant differences found between healthy controls and those with adenoid hypertrophy when using the median test in the following areas of well-being: social functioning resulting from behaviour or emotional state (REB), pain and discomfort (BP), and family cohesion (FC). However statistically significant differences were found in the deterioration of the sick children’s well-being as follows: assessment of the child’s general condition (STAND), physical fitness (PF), impact of physical health on limitations in social functioning (RP), behaviour (BE), mental health (MH), self-esteem (SE), general health perception (GH), impact of children’s health status on parents’ emotions (PE), limitations of parents’ free time due to the child’s health (PT), and restrictions on joint family activities (FA). 

## 5. Discussion

Mean values of HRQL found in the literature ranged from 53.2 to 83.0 for patients with adenoid hypertrophy [11,12,13]. These studies mainly used specific questionnaires to investigate the impact of the treatment method on the quality of life of children with this condition. They also observed a relationship between the extent of the limitations in family activities and the physical activity of children with adenoid hypertrophy. They have also shown limitations in the quality of life in areas of general health, behaviour, and emotions expressed by parents in relation to their child’s health. A strong correlation was found between reduced well-being in these areas of life and the extent of limitations in their parents’ free time in those with adenoid hypertrophy. The study author has found papers from the literature that have investigated the quality of life in ENT patients suffering from sleep apnea syndrome. An increased school absenteeism was observed in children with such sleep disorders, which, however, returns to normal after the applied treatment [12,14,15].

The present study shows that mean values for the quality of life in children with adenoid hypertrophy ranged between 59.74 and 92.63. The lowest value for general health perception (GH) was 59.74. Low values were also seen in child behavior (BE) 71.29, mental health quality (MH) 70.20, the child’s self-esteem (SE) 78.47, effect of health status on the parents’ emotions (PE) 67.57, limitations to joint family activities (FA) 76.07, and family cohesion (FC) 71.93. However, the highest indicators noted were in physical fitness (PF) 92.63 and social functioning resulting from behaviour or emotional state (REB) 91.64. A statistically significant deterioration in the quality of life was found in children with adenoid hypertrophy compared to controls in the following areas: assessment of current health, physical fitness (PF), impact of physical health on deteriorating social functioning (RP), (BE), mental health (MH), self-esteem (SE), general perception of health (GH), impact of the child’s health on parents ‘emotions (PE), limitations of parents’ free time resulting from the child’s health status (PT), and limitations in joint family activities (FA). The results have shown that tonsil hypertrophy in children appears to not only reduce the quality of life in assessment of the child’s general condition (STAND), physical fitness (PF), impact of physical health on limitations in social functioning (RP), behaviour (BE), mental health (MH), self-esteem (SE), general health perception (GH), impact of children’s health status on parents’ emotions (PE), limitations of parents’ free time due to the child’s health (PT), and restrictions on joint family activities (FA), but also limits how a family functions, leading to how an entire society thus functions.

Franco et al. studied the effect of physical health on the limitations in social functioning (RP) and general sense of health (GH) in a group of children in the USA with night apnea syndrome [14]. The assessed indices allowed him to conclude that adenoid hypertrophy deteriorates the quality of life to a moderate to significant degree, even though the questionnaire differed in details from the one in our study. Similar observations were made by Mitchell et al. on a smaller scale (n = 60 children) than in our paper; however, analogous results were obtained [12]. Similar conclusions were also drawn in a study by de Serres et al. [16]. The study by Shteinberg YH gives valuable conclusions on supplementing the clinical examination of patients by assessing their quality of life [17].

General purpose questionnaires are used for large populations with a variety of pathologies that have enabled mutual comparisons to be made between each obtained result, irrespective of whether the investigated subjects were healthy or suffering from any ailments and whether study groups differed in size. General purpose questionnaires, as research tools, are however not useful for assessing discrete changes occurring in any given individual. Specific HRQL research questionnaires have been created for any given specific issue to be investigated. They are more sensitive in detecting deviations occurring within time, so they are used in examining the effectiveness of treatment or progression of lesions. They are not, however, suitable for evaluating patients with several diseases. The general questionnaire we used for assessing the quality of life in our study has enabled us to assess the quality of life in children suffering from tonsil hypertrophy in relation to healthy children. Nevertheless, this questionnaire limits the possibilities of linking a patient’s clinical condition with their quality of life; and for this purpose, specific questionnaires will be used in the future.

## 6. Conclusions

The study enabled a comparison to be made between those children with tonsil hypertrophy and healthy children using the questionnaire CHQ-PF-50, which found that the greatest reduction in well-being occurs in the areas of behaviour and general perception of health and mental health in children suffering from adenoid hypertrophy, which were found to be significantly lower than in healthy controls, as determined by the median test method from the questionnaire data. We suggest that further like studies be performed in other parts of the world on a greater number of children and test centres to provide more statistical power and confidence to the present study outcomes. The questionnaires used are relatively straightforward. Quality of life should be an element of the patient’s clinical examination.

## Figures and Tables

**Table 1 ijerph-18-08935-t001:** Summary statistics—continuous variables for the control group.

Control	N	Mean	Std. Dev.	Min.	Q25	Median	Q75	Max.	Skewness	Kurtosis
STAND	102	3.78	0.86	1.00	3.40	4.40	4.40	5.00	−0.88	0.11
PF	102	97.11	5.17	77.78	94.44	100.00	100.00	100.00	−2.01	3.69
REB	102	96.51	7.49	66.67	100.00	100.00	100.00	100.00	−2.29	4.85
RP	102	96.24	9.92	50.00	100.00	100.00	100.00	100.00	−2.75	7.09
BP	102	85.39	16.75	10.00	70.00	90.00	100.00	100.00	−1.37	2.94
BE	102	79.19	11.15	55.00	71.67	80.83	89.17	100.00	−0.41	−0.60
MH	102	79.80	13.62	30.00	70.00	80.00	90.00	100.00	−0.87	0.52
SE	102	80.19	14.07	37.50	70.83	83.33	91.67	100.00	−0.91	0.59
GH	102	75.41	13.12	29.17	68.33	76.67	85.00	100.00	−1.08	1.52
PE	102	77.21	14.21	41.67	66.67	75.00	91.67	100.00	−0.30	−0.47
PT	102	90.41	11.60	66.67	88.89	88.89	100.00	100.00	−0.97	−0.30
FA	102	85.29	12.90	50.00	75.00	89.59	95.83	100.00	−0.78	−0.19
FC	102	66.57	18.66	0.00	60.00	60.00	85.00	100.00	−0.76	0.77
Age	102	10.58	3.55	5.00	8.00	9.00	13.00	18.00	0.80	−0.41

**Table 2 ijerph-18-08935-t002:** Summary statistics—continuous variables for the test group (adenoid hypertrophy).

Control	N	Mean	Std. Dev.	Min.	Q25	Median	Q75	Max.	Skewness	Kurtosis
STAND	101	3.58	0.76	1.00	3.40	3.40	4.40	5.00	−0.85	1.59
PF	101	92.63	11.44	44.44	88.89	100.00	100.00	100.00	−2.33	6.46
REB	101	91.64	16.13	22.22	88.89	100.00	100.00	100.00	−2.38	5.79
RP	101	88.61	19.99	0.00	83.33	100.00	100.00	100.00	−1.99	4.11
BP	101	83.17	22.00	30.00	60.00	100.00	100.00	100.00	−0.85	−0.87
BE	101	71.29	16.37	25.83	60.00	72.50	83.33	100.00	−0.53	0.06
MH	101	70.20	18.19	10.00	60.00	75.00	85.00	100.00	−0.92	0.92
SE	101	78.47	14.72	37.50	66.67	79.17	91.67	100.00	−0.51	−0.28
GH	101	59.74	17.53	8.33	47.50	60.00	72.50	89.17	−0.40	−0.25
PE	101	67.57	22.20	0.00	58.33	66.67	83.33	100.00	−0.48	−0.18
PT	101	81.85	19.23	0.00	77.78	88.89	100.00	100.00	−1.84	5.15
FA	101	76.07	19.91	12.50	66.67	79.17	91.67	100.00	−0.84	0.40
FC	101	71.93	17.62	30.00	60.00	60.00	85.00	100.00	−0.43	−0.20
Age	101	8.62	2.88	5.00	6.00	8.00	11.00	17.00	0.77	−0.12

**Table 3 ijerph-18-08935-t003:** A comparison of children suffering from adenoid hypertrophy (test group) with controls.

	Control Means	Test Means	Control Medians	Test Medians	Pt. Medians
**STAND**	3.78	3.58	4.40	3.40	**0.0035**
**PF**	97.11	92.63	100.00	100.00	**0.0185**
REB	96.51	91.64	100.00	100.00	0.1030
**RP**	96.24	88.61	100.00	100.00	**0.0041**
BP	85.39	83.17	90.00	100.00	0.0581
**BE**	79.19	71.29	80.83	72.50	**0.0002**
**MH**	79.80	70.20	80.00	75.00	**0.0010**
**SE**	80.19	78.47	83.33	79.17	**0.0295**
**GH**	75.41	59.74	76.67	60.00	**0.0000**
**PE**	77.21	67.57	75.00	66.67	**0.0364**
**PT**	90.41	81.85	88.89	88.89	**0.0049**
**FA**	85.29	76.07	89.59	79.17	**0.0059**
FC	66.57	71.93	60.00	60.00	0.1401

The bold in the table was adopted as being significant statistical.

## Data Availability

The data presented in this study are available on request from the corresponding author.

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
