# Peer review of "Health-Related Quality of Life Assessed in Children with Adenoid Hypertrophy"

_ijerph, 2021, doi:10.3390/ijerph18178935_

Round 1
Reviewer 1 Report
At present, nasal endoscopy represents the first choice in the evaluation of adenoid hypertrophy, replacing radiology.
The degree of adenoid hypertrophy in both the test and control groups was not specified in the study and the related investigation method was not defined.
I invite you to specify whether, among the exclusion criteria, there were allergic rhinitis or other pathologies responsible for nasal obstruction in childhood (extra adenoid hypertrophy)
Author Response
I begin by thanking you for the opportunity to submit our work to such a prestigious international journal. We thank the reviewers for considering our study. We found the reviewers' comments to be invaluable and inspiring. For this reason, we are grateful to have received advices that have allowed us to improve our work and we hope to have satisfied all your comments.
Both the degree of hypertrophy and the related investigative methods have been inserted into the ‚Intraduction nad Results’ section, which thus reads in the Introduction „ Assessment of adenoid hypertrophy was based on the ratio of tonsil size to the posterior nostrils and their narrowing in the study test group, expressed as a percentage. Three degree categories for hypertrophy were used; mild <60% (without clinical symptoms), moderate (60%-80%) and severe (80%-100%)”. In the Results „All children in the study test group had either moderate or severe adenoid hypertrophy. There were no statistically significant relationships between the degree of tonsil hypertrophy with reductions in any of the areas examined on the quality of life. This may have been related to the type of test we used (a general all-purpose test)”.Exclusion criteria for the control group included chronic diseases, including tonsil hypertrophy.
In the exclusion criteria, we now provide a mention in the ‘Study materials’ section on specified allergic rhinitis and have also taken into account other pathologies responsible for nasal obstruction such as chronic sinusitis and nasal septum deviation This now reads „Exclusion criteria were ages under 5 or over 18 years, acute childhood diseases, chronic childhood diseases(also included other chronic diseases such as allergic diseases, chronic sinusitis and nasal septum deviation) and incompletely filled-in questionnaires.”
Reviewer 2 Report
Abstract – there is no Results section.
Keywords – add 'Pediatric'
Introduction
- Line 33: "A suitable questionnaire was selected in order to fullfill the study aims. " Why is this sentence is in the past tense?
- Line 45: "are important" should ne "have an important"
- Line 47: " in so predisposed individuals.ors" should be " in predisposed individuals."
- Line 49: I am not sure the word "slats" should be used in this setting.
- Line 66: Suggested reference that shows the limited role of lateral x-rays (Adenoid Obstruction Assessment in Children: Clinical Evaluation Versus Endoscopy and Radiography. Kugelman N, et al. Isr Med Assoc J. 2019 Jun;21(6):376-380. PMID: 31280504).
Study materials
- Line 81: "Admission cirteria" should be "Inclusion criteria"?
- Line 87: question mark should be erased.
- Line 98: "comprising" should be "comprising of".
- Line 100: "depend" should be "depends".
- Line 106: "is then" should be "was then".
- Line 111: "and. Values" should be "and values".
Variables
- Line 123: - The terms should be used in a sentence.
Results
- Tables footnotes should consist of all abbreviations used.
- 2. There is comparison of the control to the study group. Are the groups similar or different in their composition of age and gender?
3. Table III: There is no report of 95%CI of the results so it is hard to draw any meaningful conclusions.
Discussion
- Line 186. More recent references are available. For example: Translation and cultural adaptation of the Hebrew version of the Pediatric Sleep Questionnaire: a prospective, non-randomized control trial. Shteinberg YH et al. Sleep Breath. 2021 Mar;25(1):399-410. PMID: 32394315.
2. The author should relate in their discussion whether the differences in QoL between the two groups have clinical significance. For example, I am not sure what does a difference of 3.78 vs. 3.58 in STAND mean? Or 80.19 vs. 78.47 in SE?
3. A limitation and strengths section is missing from the Discussiom.
Conclusion
- The conclusion should be better rephrased stating the tool used, the control group etc.
2. Is there any place for further studies?
References
- I guess references 3 and 4 are citations from a book chapter. If so – they should be written differently.
2. Same with reference 8.
Author Response
I begin by thanking you for the opportunity to submit our work to such a prestigious international journal. We thank the reviewers for considering our study. We found the reviewers' comments to be invaluable and inspiring. For this reason, we are grateful to have received advices that have allowed us to improve our work and we hope to have satisfied all your comments.
Abstract – apologies for the oversight. We now include a ‘Results’ part to the abstract as follows:
There were 101 filled out questionnaires for the test group (54 girls and 47 boys), mean age 8.62 years (ranging 5-17), whilst 102 questionnaires for the controls (50 girls and 52 boys), mean age 10.58 years (ranging 5-18). Insignificant differences were found between social functioning resulting from behaviour or emotional state (REB), pain and discomfort (BP) and family cohesion (FC).
Keywords: ‘Paediatric’ has now been added
Introduction
1. Changed as shown
2. Changed to ‘have an important’
3. Changed as shown.
4.Changed to „folds”
5. Suggested reference has now been added.
Study materials
1. Changed as shown.
2. Changed as shown.
3. Changed as shown.
4. Changed as shown
5. Changed as shown.
6. Changed as shown.
Variables
1. This has been changed to ‘These were subjected to analyses of medians, averages and correlation.’
Results
1. Footnotes have been changed accordingly-a shortcut list has been created
2.
The comparisons are shown in the main ‘Results’ section, including age & gender. The summarised results were mistakenly omitted from the ‘Abstract’ section but have now been inserted, where comparisons are now made clear, albeit in summary.
3.
The study aimed to answer the question whether the observed differences in 13 variables between the tonsil hypertrophy group and controls were statistically significant, keeping in mind that it is impossible for our variables to exist in the range from -∞ to + ∞. We chose the median test as it is the appropriate one to use distributions of our variables which thus precluded the use of any parametric tests that would lead to serious statistical errors being committed. We therefore rejected using parametric testing and instead used non-parametric tests, can be used for variables with any types of distribution. The median test is such a test, where the test statistic has a Chi-square distribution with one degree of freedom (df = 1) at a p=0. 05 level of significance. Using the median test we have thus the statistical confidence to state whenever differences are significant which are so marked in red.
Discussion
The discussion has been changed to incorporate the reviewers comments
Conclusions
The conclusion has been changed to incorporate the reviewers comments.
Reference
The reference has been changed to incorporate the reviewers comments.
Round 2
Reviewer 1 Report
It's ok
Reviewer 2 Report
No further comments